# Aluminum Nitride Thin Film Piezoelectric Pressure Sensor for Respiratory Rate Detection

**DOI:** 10.3390/s24072071

**Published:** 2024-03-24

**Authors:** Maria Assunta Signore, Gabriele Rescio, Luca Francioso, Flavio Casino, Alessandro Leone

**Affiliations:** The National Research Council, Institute for Microelectronics and Microsystems (CNR IMM), Via Monteroni, 73100 Lecce, Italy; mariaassunta.signore@cnr.it (M.A.S.); lucanunzio.francioso@cnr.it (L.F.); flavio.casino@cnr.it (F.C.); alessandro.leone@cnr.it (A.L.)

**Keywords:** piezoelectric pressure sensor, aluminum nitride (AlN), breathing monitoring, wearable device, flexible electronics, healthcare

## Abstract

In this study, we propose a low-cost piezoelectric flexible pressure sensor fabricated on Kapton^®^ (**Kapton**™ **Dupont**) substrate by using aluminum nitride (AlN) thin film, designed for the monitoring of the respiration rate for a fast detection of respiratory anomalies. The device was characterized in the range of 15–30 breaths per minute (bpm), to simulate moderate difficult breathing, borderline normal breathing, and normal spontaneous breathing. These three breathing typologies were artificially reproduced by setting the expiratory to inspiratory ratios (E:I) at 1:1, 2:1, 3:1. The prototype was able to accurately recognize the breath states with a low response time (~35 ms), excellent linearity (R^2^ = 0.997) and low hysteresis. The piezoelectric device was also characterized by placing it in an activated carbon filter mask to evaluate the pressure generated by exhaled air through breathing acts. The results indicate suitability also for the monitoring of very weak breath, exhibiting good linearity, accuracy, and reproducibility, in very low breath pressures, ranging from 0.09 to 0.16 kPa. These preliminary results are very promising for the future development of smart wearable devices able to monitor different patients breathing patterns, also related to breathing diseases, providing a suitable real-time diagnosis in a non-invasive and fast way.

## 1. Introduction

The development of wearable health monitoring systems is nowadays considered a winning strategy to meet the needs for portable devices able to provide reliable information to support the early detection and medical diagnosis of health conditions. Flexible pressure sensors play a crucial role in this field [1,2]. They are devices capable to transduce external pressure into an electrical signal, providing physiological parameter detection in real-time. Pressure sensors are usually classified by their transducing mechanisms, i.e., piezoresistive [3], capacitive [4], and piezoelectric [2]. Piezoresistive sensors exhibit a significant temperature dependence response and they require a large amount of power to work. Capacitive ones have high sensitivity and simple geometry, but they require a high voltage to operate. Piezoelectric pressure sensors, especially those based on III-N thin films, produce detectable charge signals by themselves after applied mechanical pressure, due to the piezoelectric effect, that is, the induction of electric charges in response to an applied mechanical strain, widely encountered in many natural materials that do not have a center of symmetry. These classes of sensors are considered very attractive based on their several desirable attributes over the other sensor types, such as high sensitivity, fast response time, wide dynamic range, simple fabrication way, affordable costs, easy output signal acquisition, no need of extra poling, biocompatibility, and self-powering [5,6]. One of the most investigated emergent application areas of wearable pressure sensor deals with breathing condition monitoring [7], being the respiration rate one of the most highly informative and predictive vital parameters for a large range of medical diagnoses [8]. It has also been demonstrated that the respiratory rate is a useful and reliable biomarker for the detection of some respiration diseases, like COVID-19 [9]. Modern technologies to measure breathing rate include pulse oximetry [10], electrocardiography (ECG)-derived method [11], RF-based approach [12], and the auscultation of the chest counting breaths for one minute by the aid of watches or a timer, with the possible imprecision and inaccuracy of the measurements [13,14]. Polysomnography is the gold standard method used in laboratories to measure the respiration rate during sleep [15,16]. Commercial wearable devices are mainly based on the photoplethysmography technique, performing measurements at the skin surface, such as wrist, earlobes, or fingertips [17]. Despite the benefits of these approaches, some drawbacks should be considered: each one requires high-cost wearable sensors, or uncomfortable probes linked to the patient’s body, or professional systems used by trained staff [18]. As a result, there is a widely recognized clinical need for real-time breath monitoring devices with specific characteristics: ease of use, automatic, repeatable, and reliable response, small size and ease of wearing, cost-effectiveness, low power consumption, high accuracy, reduced artifacts caused by patient movement during breath monitoring [19,20]. Recent literature reports about the design and fabrication of flexible piezoelectric pressure sensors for human breath monitoring due to the above well-known properties, in combination with the high electromechanical coupling and quality factor, are required for the detection of fast changes in respiration flow [21,22]. The reference values considered in these works to describe a normal respiration rate for an adult are approximately 12–20 breaths per minute (bpm); values above 20 or below 12 bpm describe a critical clinical condition [23]. For example, a respiratory rate above 27 bpm has been shown to be predictive of cardiopulmonary arrest [24]. Among ceramic piezoelectric materials used for pressure sensors fabrication, aluminum nitride (AlN) is one of the most considered due to its interesting properties suitable for this application. It works in a stable way in high-temperature environments, keeping its piezoelectric characteristics at temperatures up to 1150 °C [25]. Moreover, AlN does not require electric-thermal poling such as ferroelectrics, being naturally piezoelectric, and it is a lead-free environmentally friendly ceramic material compared to lead zirconate titanate (PZT), which suffers the presence of hazardous lead (Pb) despite its excellent piezoelectric properties [26]. The sensitivity of AlN-based piezoelectric pressure sensors does not decrease at high temperature as for other materials (i.e., GaN): being the widest bandgap (E_g_) piezoelectric material (E_g_ = 6.2 eV), AlN was demonstrated as the best-performing temperature pressure sensor, and is particularly promising for applications in harsh environments, able to overcome the sensitivity limitation with temperature [27]. To the best of our knowledge, AlN is proposed for the first time as an active material for a lead-free flexible piezoelectric pressure sensor fabrication to monitor respiration rate through the air flow incoming on active area during the respiratory act flow. The main aim is the detection of different respiratory states, included breathing anomalies. Only one other example is reported the literature [28], though it is not a recent study, where an AlN-based pressure sensor was placed in contact with a the chest of a lying subject to measure pressure fluctuations due to respiration and heartbeat movements, but not caused by breathing flow. Our device was initially tested by varying the expiration/inspiration times ratios (E:I) in artificial atmospheres (dry air), simulating three typologies of breathing, i.e., normal spontaneous breathing, borderline normal breathing, and moderate breathing difficulty [29]. Our results showed that it was able to reproduce different breath patterns by exploiting the direct piezoelectric effect with a low response time and promising reproducibility and linearity. Successively, the device was characterized in real conditions consisting of monitoring pressure patterns produced directly by respiratory acts, showing an excellent capability to detect very low-pressure signals attributable to breathing acts coming from different individuals. The fabricated flexible pressure sensor represents a good candidate as a smart wearable sensor designed for breath signal monitoring, providing diagnosis in a non-invasive, fast, and real-time way. It is completely biocompatible due to the stated biocompatibility of the involved material, i.e., Kapton^®^ substrate [30], AlN [31] and titanium (Ti) [32]. Moreover, the sensor microfabrication process makes it suitable for mass production as a low-cost wearable sensor for biomedical applications.

## 2. Materials and Methods

The piezoelectric pressure sensor has been fabricated using a stacked-layers approach, where the proprietary AlN piezoelectric film (500 nm thick) is inserted in between two Ti electrode layers (150 nm thick). The process starts with Kapton^®^ substrate preparation; then, a photoresist layer is spin-coated and patterned for three different consecutive steps to realize the Ti/AlN/Ti multilayer by RF magnetron sputtering deposition. More details about thin films deposition and the device fabrication process can be found in [33]. Figure 1 shows the sketch of the device where the bottom and top electrode diameter sizes are indicated (Figure 1a), and the photo of the fabricated piezoelectric pressure sensor on flexible Kapton^®^ substrate (Figure 1b).

The morphology, the structure, and the piezoelectric response of the AlN active layer were investigated. Surface morphology has been characterized by Atomic Force Microscopy (AFM) measurements (Nanosurf-Core AFM) in air in dynamic mode. The structural properties of the investigated sample deposited on Kapton^®^ have been investigated by X-ray diffraction (XRD) by using the Cu-Kα radiation and scanning angle of 2θ = 10–80°. In order to characterize the piezoelectric response of AlN thin film, a Piezo Force Microscopy (PFM) instrument (CoreAFM from Quantum Design, Italy) was used to evaluate the effective longitudinal piezoelectric constant d_33_. By this technique, the local sample vibrations caused by an AC signal applied between the conductive tip and the bottom electrode of the sample were detected. A chromium-platinum coated conductive tip (radius of 25 nm, MULTI75-EG, BudgetSensors, Innovative Solutions Bulgaria Ltd.) was used and the AC bias voltage applied to the tip ranged from 2 V to 5 V. The frequency was set at 3 kHz to prevent mechanical oscillation of the tip at its resonant frequency (75 kHz). The PFM measurements have been performed considering only the vertical displacement of the thin film and the piezoelectric coefficient d_33_ was calculated from the linear regression of vibration amplitude versus applied AC voltage. The output signal from the photodiode has been calibrated by a calibration sample of lithium niobate (LiNbO_3_) consisting of periodically poled structures with a known piezoelectric coefficient. To characterize the sensor, two different experimental setup configurations were realized. The first one, shown in Figure 2a, is used to characterize the pressure sensor in an artificial atmosphere, essentially made up of dry air certified Pure Gas, to recreate finely controllable and reproducible conditions.

The experimental setup consists of a controlled atmosphere cell (3) housing both the Device Under Test (DUT) sensor (4) and the commercial fully calibrated piezoresistive silicon pressure sensor taken as reference (SSCDRRN025MDAA3) (5). Through an adjustable gas decompression station (1), the dry air is blown in the box through an inlet point with a fine and accurate pressure regulation in the range of interest (2), and then it is released through an outlet hole. In order to vary the frequency of air intake, in accordance with typical breath timing, a PCB equipped with relays (DC 8–36 V four-channel multifunction relay control module, KR-122-4-V1.0) was used (7), capable of driving several air valves in a programmable sequence. The box allows to reach inside air volumes comparable to those present during breathing, while the outlet hole has been configured in such a way to simulate a medical mask exhaust valve. A conductive shielding paint has been deposited inside the box to reduce EMI effects. The pressure was evaluated by the commercial pressure sensor positioned next to the DUT. The reference sensor has an operating range between 0 and 20 kPa, thus it is compatible with the breath values of interest for this investigation. The second configuration is used instead to characterize the piezoelectric sensor in real operative conditions consisting of pressure modulations generated by respiratory acts. Figure 2b shows the schema of the setup where the adjustable gas decompression station has been replaced by the Personal Protective Equipment (PPE) mask with an activated carbon filter. The electrical signal generated by the DUT sensor under simulated and real breathing flow actions is acquired by a voltage amplifier (DLPVA-100-F Series—FEMTO). The output values of the voltage amplifier were acquired and stored via a digital oscilloscope (GDS-2204A), and then processed in the programming language and numeric computing environment software Matlab R2023b. In particular, the data was filtered to mainly reduce environmental noise.

The experimental parameters simulate different breathing conditions, taking into account the expiratory (E) to inspiratory (I) times ratio (E:I) [29], set at 3:1, 2:1, 1:1. Specifically, E:I = 2:1 simulates a borderline normal breathing act, E:I = 1:1 reproduces a moderate breathing difficulties and finally E:I = 3:1 simulates a normal spontaneous breathing, considering that the average respiratory rate of an adult ranges from 12 to 20 bpm during the rest condition. The simulated breathing conditions are reported in Table 1.

The setup configuration described in Figure 2a was mainly exploited to verify the capability of the sensor to reproduce exactly different breathing patterns. Once this capability was established, the DUT suitability for the detection of very small pressure variations, similar to breathing pressures at normal conditions, was investigated through human respiratory acts.

## 3. Results and Discussion

### 3.1. Morphological, Structural, and Piezoelectric Characterization of AlN Thin Film: AFM, XRD and PFM Analyses

Figure 3 shows the AFM scan of AlN thin film deposited on Ti seed layer on Kapton^®^ substrate. The root means square roughness (R_q_) evaluated on ten scans is equal to (20 ± 0.2) nm.

In the inset of Figure 3, the XRD spectrum of the same film is shown. The 2θ angle range has been reduced (30°–50°) in the interval where peaks of interest have been detected. The peak around 2θ = 36° is attributable to the reflection from (0002) planes of hexagonal wurtzite structure of AlN while Ti(0002) reflection from seed layer is located at 2θ = 38.5°. AlN film preferentially oriented along c-axis orthogonal to the substrate is required to experience piezoelectricity by the nitride. The piezoelectric response of the active material was investigated by PFM. Figure 4 shows the displacement of the film versus the AC voltage applied between the sample and the tip. The slope of the linear fit provides the value of the piezoelectric coefficient d_33_ resulted to be equal to 1.3 pm/V. On the right, the representative PFM images of amplitude and phase obtained at V_AC_ equal to 2 V, which provides quantification of displacement intensity and polarization direction, respectively.

The characterized AlN thin film also exhibits the biocompatibility property, as already reported in [31], which is an important property for the wearability suitability.

### 3.2. Piezoelectric Pressure Sensor Characterization

The piezoelectric sensitivity of the sensor (i.e., the amount of charge/voltage per unit force or pressure) can be used to describe the performance of such sensor. Figure 5 depicts the trend of the piezo-potential generated by the flexible sensor by varying the pressure applied to the active nitride area in the typical interval of breath pressure; as reported in the literature, at-rest breathing generates a modest pressure swing of ±0.4 kPa to create inhalation and exhalation flow [34].

The evaluated sensitivity is equal to 15.5 mV/kPa with an accuracy > 99%. This result is fully consistent with recent literature about flexible piezoelectric pressure sensors specifically employed for breath monitoring. It is higher than the sensitivity found J. Hu et al. equal to 12 mV/kPa by fabricating a PVDF-based flexible device, and also very close to the 19 mV/kPa sensitivity measured by the same authors in low pressures range (within 500 Pa) [35]. In the inset of Figure 5, the output signal of the piezoelectric sensor was plotted versus increasing pressure values and successively decreasing pressure values in the same experiment, with the aim to evaluate hysteresis. As is clearly observable, it exhibits the same responses in both directions, confirming a low hysteresis which guarantees a reliable repeatability and stability of the sensor’s output. In this preliminary study, we studied the pressure sensor response in the 15–30 bpm range to obtain respiratory information. Figure 6 shows the sensor output at different frequencies of expiration/inhalation in the selected interval, by using the experimental setup described in Figure 2a. The acquired profiles clearly reflect the subject’s actual breathing mode.

As can be observed, the pressure pulse wave hitting the pressure sensor is not an exact sinusoidal wave due to fluid turbulence which affects the shape of the sensor response. The response time can be accurately evaluated from these waveforms as the rise time between 10% and 90% of the ascending edge of the output response. It resulted to be about 35 ms, as shown in Figure 6 for a waveform filtered through a low-pass filter implemented in Matlab used for the calculation of the response time of the pressure sensor.

This rapid responsive characteristic of the piezoelectric sensor to the pressure variations is a very relevant aspect for an accurate estimation of the respiration rate. This guarantees its applicability for the continuous monitoring and detection of sudden changes in breath rhythm to assess the health state of a patient in real-time. Furthermore, this result is mainly supported by the peculiarity of piezoelectric pressure sensors which are suitable only for dynamic pressure measurements through the generation of a voltage proportional to the mechanical deformation caused by the applied pressure. As can be seen in Figure 7, the recovery time is not so trivial. This behavior could be ascribed to the experimental conditions not being completely optimized, where the emptying of the controlled atmosphere cell during the simulated breathing act is not so fast, causing the persistence of membrane deformation. An alternative approach to monitor respiration was followed by placing the piezoelectric pressure sensor attached on a mask [36] as was explored by characterizing the DUT sensor with the experimental setup previously described in Figure 2b. In line with the previously discussed experiments in an artificial atmosphere, initially, the sensor’s ability to distinguish different breathing modes was tested. A preliminary data acquisition campaign was carried out by considering different respiratory abnormalities of the potential monitored patient. Figure 8 shows some signals acquired from two users. Hyperventilation and coughing conditions were sequentially simulated during the experiments. Figure 8a,c shows the data acquired by the DUT sensor, and Figure 8b,d shows the data read with the commercial sensor used as a reference.

As is clearly observable from the figures, although the waveform of the two sensors is different (due to their different working principle), both measure a similar mean respiratory rate. Moreover, in the presence of coughing, the waveform shape changes (spikes appear), so it makes it possible to detect this event through appropriate software techniques. The aforementioned tests show that the DUT sensor can discriminate different respiratory conditions with good accuracy, even in real-world settings. Once the capability of the sensor to discriminate different breathing patterns, was checked, the breath was monitored across the activated carbon filter mask in very low pressures, ranging from 0.09 to 0.16 kPa, or similarly, at a range well below the normal breathing range. This experimental choice aimed to verify the correct working capability of the fabricated flexible pressure sensor in severe conditions, like very feeble breath. The DUT sensor showed a very good linearity equal to 0.974%, as depicted in Figure 9, with an expected lower sensitivity in the range of extremely low breath pressures (compared to the previous tests), but motivating and appropriate for the application in very severe respiration conditions.

These preliminary promising results confirm the suitability of the fabricated flexible pressure sensor for wearable biomedical diagnostics. Moreover, owing to its key properties, like small size and flexibility, the single piezoelectric sensor can be easily placed on any body’s surface where mechanical deformation occurs, making its use even more versatile for wearable devices. In addition, the sensors can be arranged as arrays in a very small volume, making possible the simultaneous evaluation of external pressures coming from different points of the body. Table 2 reports a comparison of our results with the most recent literature ones about wearable flexible piezoelectric pressure sensors employed for breathing analysis where the DUT active area deformation is due exclusively to the breathing flow hitting the sensor. The low active area size of our sensor is remarkable compared to the others, which makes it a challenging candidate for easy integration in more complex systems, together with other advantages like the biocompatibility, the easiness of fabrication, the integration of piezoelectric thin films like AlN which do not require any poling process and are extremely stable when working at high temperatures.

## 4. Conclusions

A piezoelectric pressure sensor based on AlN thin film deposited on Kapton^®^ substrate has been fabricated with the aim to provide a cheap, lightweight, and easy-to-use wearable device, able to monitor respiration rate and to give evidence of respiratory disfunctions in real-time. The proposed device has a very small size compared to the most recent results found in the literature about piezoelectric pressure sensors which directly analyze the breathing flow. This feature makes its integration into a small package extremely easy, minimizing the interference with the comfort of the patient. Moreover, restrained sensor volumes favor their arrangement in array geometries to improve the device sensitivity and final performance. Even if AlN material has lower piezoelectric constants compared to the largely used PVDF one, our choice guarantees a lower cost of the final device, making it suitable for at-home breath monitoring, given that PVDF is a more expensive material affecting the sensor price [38]. The piezoelectric effect working mechanism is extremely advantageous for the application proposed in this paper: it allows for the realization of self-powered sensors which is an important requirement for the development of portable and wearable devices, overcoming the known limitations of conventional power sources such as bulky size, operational time, and rigid structure. Our experimental results demonstrated the capability of the sensor to accurately evaluate the respiration rate with very fast response time and good sensitivity, equal to 15.5 mV/kPa (0.16–0.6 kPa) and 2.51 mV/kPa (0.09–0.16 kPa). Moreover, the proposed piezoelectric pressure sensor provides an output which clearly reflects the actual breath frequencies, giving information in real time about the health status of the subject. The optimal flexibility which very thin Kapton^®^ substrate confers to the pressure sensor is advantageous for the conformability of the device on any irregularly shaped surfaces such as masks, promoting the improvement of the smart masks nowadays available in the market, for daily health monitoring, spanning from sports activity to working environments. Moreover, new designs with smart features such as compactness and self-powering, will pave the way for the development of next-generation masks made of biocompatible materials. Future works will be devoted to package fabrication, conditioning circuit design and realization to integrate with the sensor, and to algorithmic pipeline development for the automatic detection of breathing anomalies.

## Figures and Tables

**Figure 1 sensors-24-02071-f001:**
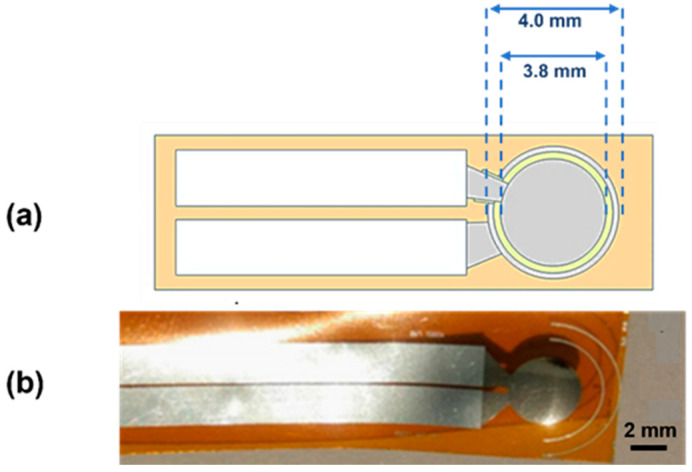
(**a**) Sketch of the designed device where the grey areas represent bottom and top Ti electrode with their diameter size (4 mm and 3.8 mm, respectively) while the yellow circle represents the piezoelectric active area; (**b**) image of the fabricated pressure sensor on Kapton^®^ substrate.

**Figure 2 sensors-24-02071-f002:**
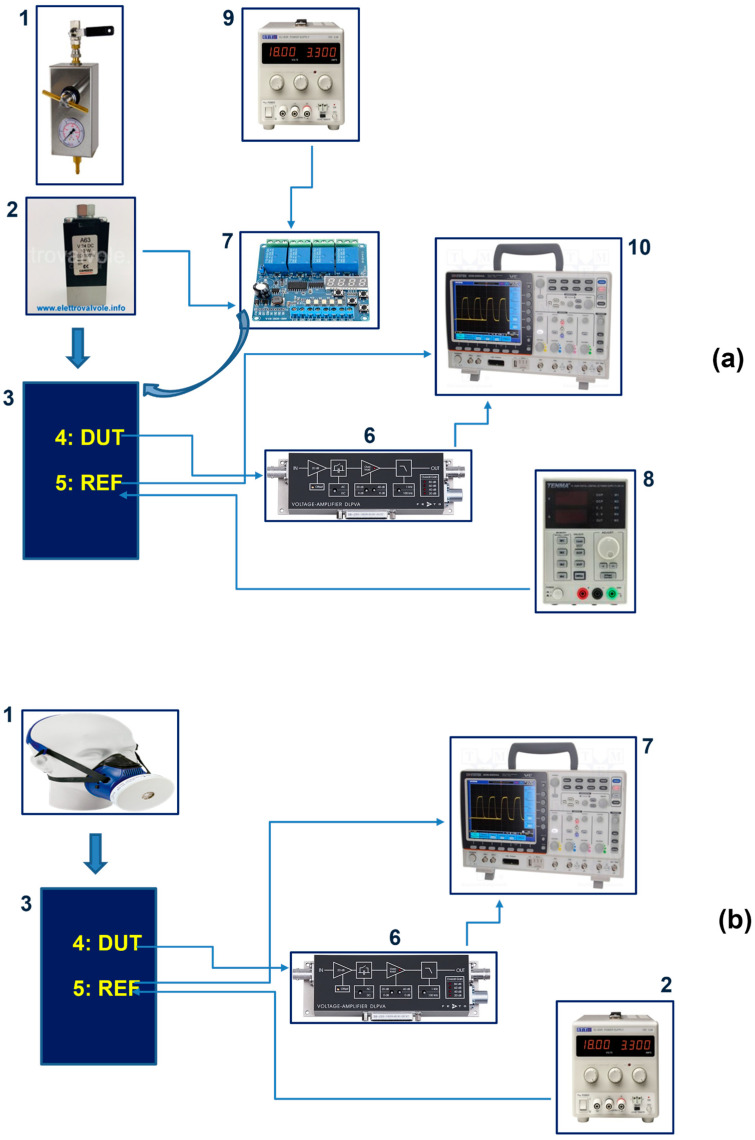
The experimental setups have been realized by using the elements reported in the image. Setup (**a**): (1) Adjustable gas decompression station; (2) Solenoid valve 24 V 5 W, NC; (3) controlled atmosphere cell housing both the commercial and the fabricated pressure sensors; (4) Device Under Test (DUT): fabricated piezoelectric sensor; (5) reference commercial sensor; (6) low-frequency voltage amplifiers FEMTO SERIES DLPVA; (7) DC 8–36 V four-channel multifunction relay control module, KR-122-4-V1.0; (8) TENMA power supply 72-10480 volt 0 → 30 V, A 0; (9) power supply Aim-TTi EL303R volt 0 → 30 V, A 0 → 3; (10) Tektronix GDS-2204A digital oscilloscope. Setup (**b**): (1) 3M PPE mask with activated carbon filter; (2) power supply Aim-TTi EL303R volt 0 → 30 V, A 0; (3) controlled atmosphere cell housing both the commercial and the fabricated pressure sensors; (4) Device Under Test (DUT): fabricated piezoelectric sensor; (5) reference commercial sensor; (6) low-frequency voltage amplifiers FEMTO SERIES DLPVA; (7) DC 8–36 V four-channel multifunction relay control module, KR-122-4-V1.0.

**Figure 3 sensors-24-02071-f003:**
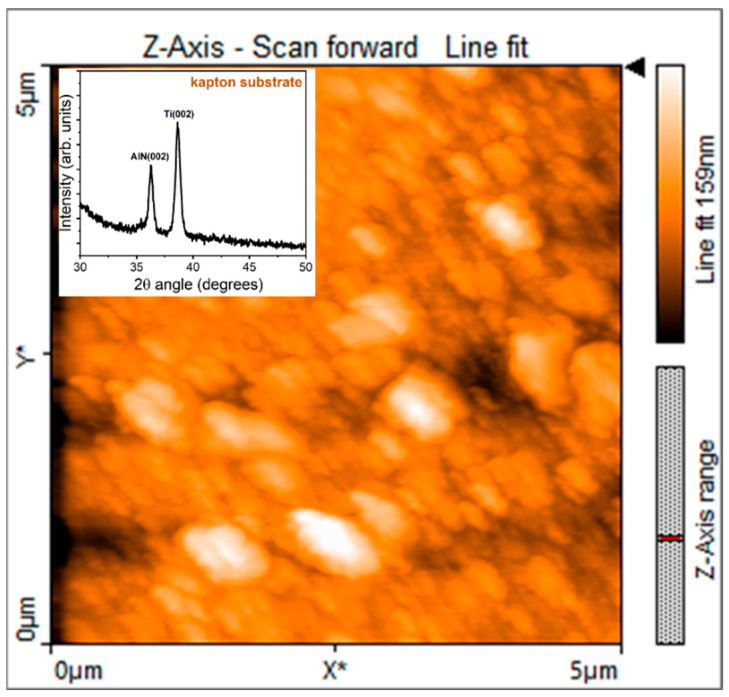
AFM image of the AlN/Ti/Kapton^®^ sample; XRD spectrum in the inset.

**Figure 4 sensors-24-02071-f004:**
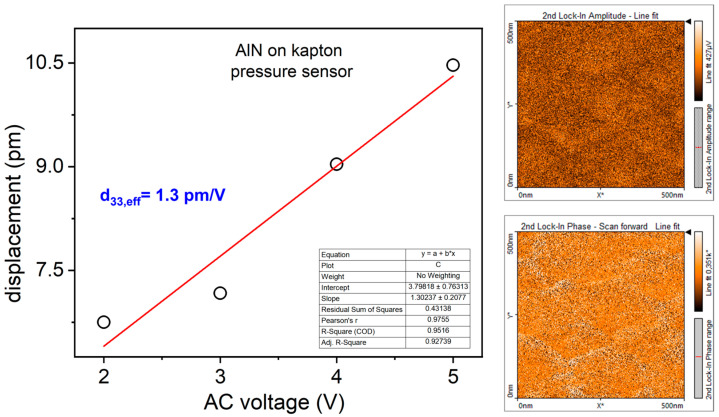
Piezoelectric coefficient of AlN thin film deposited on Kapton^®^ substrate evaluated by PFM analysis; on the right, the representative phase and amplitude images acquired at V_AC_ = 2 V.

**Figure 5 sensors-24-02071-f005:**
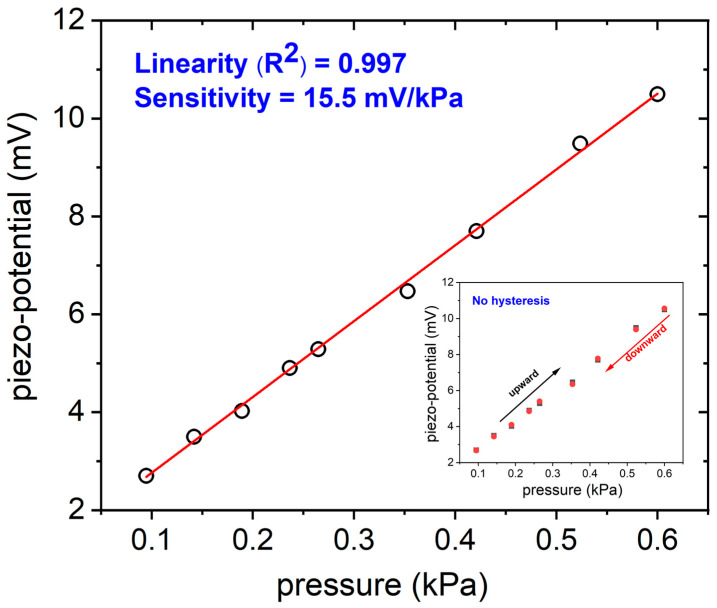
Piezo-potential generated by the piezoelectric flexible sensor in the typical breath pressures interval (hollow circles in the figure). In the inset, the hysteresis behavior of the sensor is reported (squares are experimental data acquired during upward measurements, red circles the data acquired downward).

**Figure 6 sensors-24-02071-f006:**
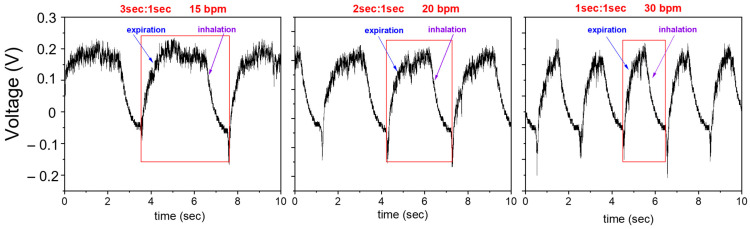
Voltage output of the piezoelectric pressure sensor at different bpm frequencies. The red frame in each figure represents the single breathing act made of expiration and inhalation.

**Figure 7 sensors-24-02071-f007:**
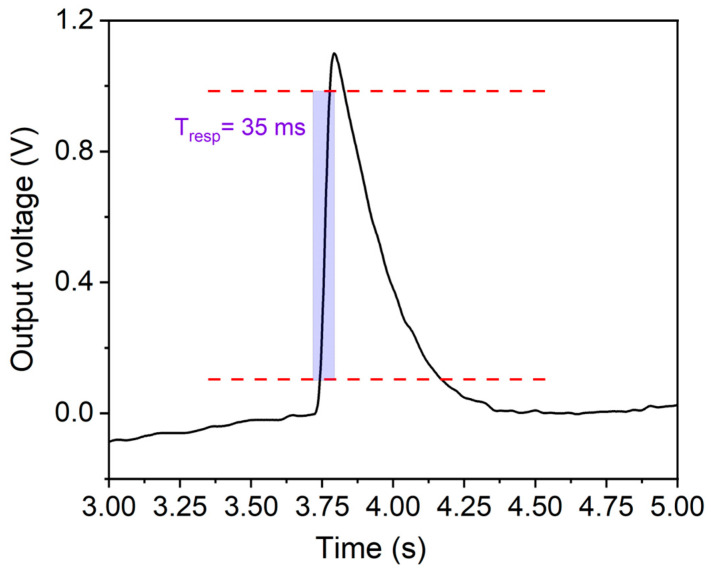
Waveform filtered through a low-pass filter implemented in Matlab used for the calculation of the response time of the piezoelectric pressure sensor. The horizontal dotted lines represent 10% and 90% of the output response in the ascending edge.

**Figure 8 sensors-24-02071-f008:**
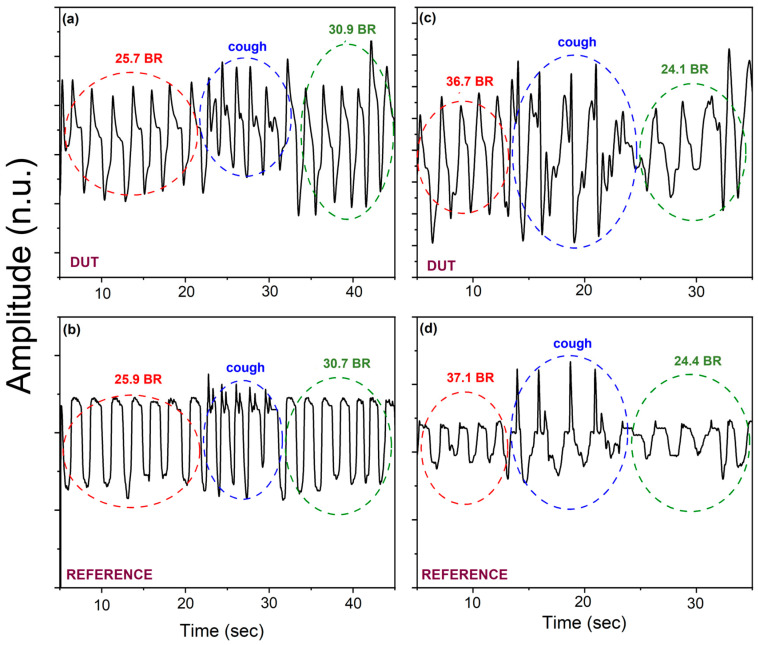
Hyperventilation and coughing conditions recorded by the DUT sensor ((**a**), user1; (**c**), user2) and by the commercial pressure sensor taken as reference ((**b**), user1; (**d**), user2).

**Figure 9 sensors-24-02071-f009:**
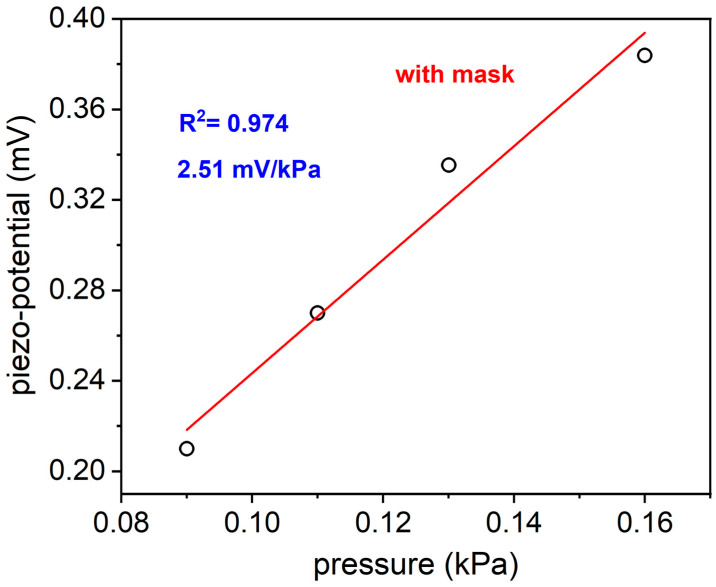
Piezo-potential generated by the fabricated piezoelectric sensor during the monitoring of very low breath pressures across the activated carbon filter mask.

**Table 1 sensors-24-02071-t001:** Simulated breathing conditions.

Case of Study	E:I	bpm
A—moderate difficult breathing	1:1	30
B—borderline normal breathing	2:1	20
C—normal breathing	3:1	15

**Table 2 sensors-24-02071-t002:** A comparison of the performance between the piezoelectric pressure sensor proposed in this work and some previously published papers taken from the most recent literature. The works referred exclusively to flexible piezoelectric pressure sensors used for respiration monitoring where the DUT active area deformation is due to breathing flow.

Material	Sensitivity	Working Range	Active Area Size (mm^2^)	References
PVDF	19 mV kPa^−1^	0–500 Pa	420	[35]
P(VDF-TrFE)/MWCNT	540 mV N^−1^	0.5–5 N	225	[37]
PVDF	4.96 V N^−1^ 940 mV N^−1^	0–0.3 N 0.3–3.8 N	180	[38]
AlN thin film	15.5 mV kPa^−1^	0.08–0.6 kPa 0.001–0.006 N	11.34	Our work

## Data Availability

The data are not publicly available due to restrictions (their containing information that could compromise the privacy of research participants).

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
