# Peer review of "Aluminum Nitride Thin Film Piezoelectric Pressure Sensor for Respiratory Rate Detection"

_sensors, 2024, doi:10.3390/s24072071_

Round 1

Reviewer 1 Report

Comments and Suggestions for Authors

In the manuscript entitled "Wearable piezoelectric pressure sensor based on biocompatible AlN thin-film", M.A. Signore et al. have proposed  a low-cost piezoelectric flexible pressure sensor fabricated on Kapton substrate by using aluminum nitride (AlN) thin film, designed for the monitoring of the respiration rate for a fast detection of respiratory anomalies.

In my opinion, the manuscript is well written and acceptable for publication in the present form.

Author Response

The authors thank the referee for the positive evaluation of the paper.

Reviewer 2 Report

Comments and Suggestions for Authors

1. Please rephrase the description of biocompatibility in the title. Your introduction lacks a proper reference and description. Please provide a recommendation regarding PZT. Below are suggested references.

- Besleaga et al., Nanomaterials 2017, 7, 394
- Berg et al., Mater. Lett. 2017, 189, 1
- Ou et al., J. Electrochem. Soc. 2007, 154, P11
- Chen et al., Adv. Funct. Mater. 2019, 29, 1903162
- Park et al., Adv. Mater. 2017, 29, 1702308

2. Authors claimed the term of “DUT” several times especially in Figure 2. However, there is no explanation. What is DUT? If this is abbreviated format, authors need to use full-word at the first time.

3. Author maintained AlN can be stable up to 1150℃. However, there is no reference to back up this claim. Suggest the references for sentence from 81st to 82nd line (1150℃).

4. In Figure 2b, it looks like the sensor is attached to the inside of the mask, so is there any effect of moisture from breathing? I believe there should be moisture effect on sensor system. Or is there something that reduces the moisture impact?

5. In the Figure 3 inset, the author measured XRD in the range of 10° - 80°, but the results are between 30° and 50°, why is this?

- And if possible, can other types of XRD measurements, such as omega and pi scans, be performed?
- Author may consider below as reference
 Kim et al., Adv. Funct. Mater. 2021, 31, 2008242

6. In Figure 4, displacement value is pm scale from PFM analysis (not nm scale), meaning that designed sensor is very sensitive to pressure from outside.

7. Please change Comma to dot (or point) in the graphs of Figure 4 to Figure 9.

Author Response

We would like to thank the reviewer for his precious suggestions necessary for improving the quality of our work. All the modifications inserted in the paper file have been yellow-evidenced to make more visible the reworked parts.

  1. Please rephrase the description of biocompatibility in the title. Your introduction lacks a proper reference and description. Please provide a recommendation regarding PZT. Below are suggested references.

- Besleaga et al., Nanomaterials 2017, 7, 394
- Berg et al., Mater. Lett. 2017, 189, 1
- Ou et al., J. Electrochem. Soc. 2007, 154, P11
- Chen et al., Adv. Funct. Mater. 2019, 29, 1903162
- Park et al., Adv. Mater. 2017, 29, 1702308

Biocompatibility is a mandatory property for wearable technology, and the authors have considered interesting to report that the proposed material meets this requirement. However, the suggestion of the referee has been accepted because the biocompatibility of AlN is not a property exploited in the application discussed in the paper, not being the sensor in contact with the human skin. For this reason, we have modified the paper title by eliminating the word “biocompatible”, following also the suggestions coming from other referees. Taking into account this revision, the authors consider useless the addition of further references about AlN biocompatibility, keeping the reference of Signore M.A. et al. that demonstrates this further characteristic of the proposed material integrated into the piezoelectric pressure sensor. About PZT, in line 88 it has been cited in comparison with AlN which is a lead-free material.

  1. Authors claimed the term of “DUT” several times especially in Figure 2. However, there is no explanation. What is DUT? If this is abbreviated format, authors need to use full-word at the first time.

Thanks to the referee for the clarification. The term “DUT” is the abbreviation form of “Device Under Test”. It has been explicated in the caption of Figure 2 (Line 148) and in the text (line 155).

  1. Author maintained AlN can be stable up to 1150℃. However, there is no reference to back up this claim. Suggest the references for sentence from 81stto 82nd line (1150℃).

As rightly suggested by the referee, a reference has been added in line 82 related to the AlN stability of its piezoelectric properties at high temperature. As consequence, all references have been re-numbered.

  1. In Figure 2b, it looks like the sensor is attached to the inside of the mask, so is there any effect of moisture from breathing? I believe there should be moisture effect on sensor system. Or is there something that reduces the moisture impact?

Thanks to the referee for this observation. Surely, future works will be devoted to the design and fabrication of the package to protect the device from moisture and its integration into a smart mask. At the moment, the sensor does not have a package, yet. For this reason, during measurements the sensing element has been properly protected from moisture by a temporary absorbent cotton coverage to reduce humidity impact.

  1. In the Figure 3 inset, the author measured XRD in the range of 10° - 80°, but the results are between 30° and 50°, why is this?

- And if possible, can other types of XRD measurements, such as omega and pi scans, be performed?
- Author may consider below as reference
 Kim et al., Adv. Funct. Mater. 2021, 31, 2008242

As rightly noted by the reviewer, the inset of Figure 3 shows the XRD spectrum of AlN thin film deposited on Kapton in 2θ angle range equal to 30° – 50° while in the “Materials and Methods” paragraph a wider range has been declared (2θ= 10° – 80°). Generally, XRD measurements are performed to scan the sample through a large range of 2θ angles, to detect all possible diffraction directions of the lattice. In this case, peaks of interest have been detected only in the shorter angle range reported in the inset of Figure 3. A sentence has been added in 3.1 paragraph to avoid misunderstanding. XRD scanning modes are different according to the information you need. The 2-theta scan, where omega is fixed at grazing angle (~0.5 or 1 or 2) and move the detector to collect the data, has advantage in polycrystalline films. The grazing angle allows for collecting diffraction signal from film, not from the substrate. Also, at grazing angle we get large exposer area compared to theta-2theta measurement. In this case for a film 500 nm-thick this XRD configuration is appropriate to have information about the (0002)-orientation of the thin films which is necessary to guarantee piezoelectric response. Psi analysis is generally performed to have information about the films stress. Unfortunately, we do not have a diffractometer able to operate in these suggested configurations. But, as already said, θ-2θ scan protocol is suitable for the aim of our work.

  1. In Figure 4, displacement value is pm scale from PFM analysis (not nm scale), meaning that designed sensor is very sensitive to pressure from outside.

Figure 4 reports the piezoelectric characterization of AlN thin film performed by Piezoresponce Force Microscopy (PFM), a powerful technique based on the reverse piezoelectric effect used to obtain the small displacement of a material forced by electric field. As output, it provides the piezoelectric coefficient as displacement in pm per V (pm/V).  Specifically, the conductive AFM tip is brought in contact with the surface of the piezoelectric material and a voltage is applied between sample and tip, generating an electric field within the material. The sample lattice can locally expand or contract according to the electric field. The vertical PFM amplitude and phase signals are related to the magnitude and the sign of the longitudinal piezoelectric coefficient, d33,eff, respectively. The typical intrinsic piezoelectric displacement of thin films is small (i.e. pm to nm range) as also correctly found for our nitride thin film. This is the typical sensitivity of the piezoelectric thin films integrated into piezoelectric sensors, i.e. piezoelectric pressure sensor, as easily can be found in literature.

  1. Please change Comma to dot (or point) in the graphs of Figure 4 to Figure 9.

As suggested, the graphs have been modified by replacing the comma with the dot. The captions have been yellow-evidenced to confirm the replacement.

Reviewer 3 Report

Comments and Suggestions for Authors

In this paper (sensors-2915828), the authors presented a piezoelectric pressure sensor based on aluminum nitride (AlN) thin film. The results are acceptable and the topic can attract a wide range of readerships. But there are many problems in the introduction, presentation, and discussion of the results. As such, a Major Revision is needed before possible publication. My specific comments are as follows:

1.      Title: “biocompatible” is not necessary to emphasize, and “AlN” needs to be spelled in full in the title.

2.      Author affiliation: The same affiliation does not need to be listed repeatedly.

3.      Introduction: (1) The abbreviations that first appear need to be spelled out in full, such as AIN “…fabrication, AlN is…”. (2) The background logic is a bit confusing. The story should be enhanced. The research on respiratory detection sensors needs to be summarized, such as pressure, strain, and humidity sensors. The more recent research progresses of advanced flexible pressure and respiration sensors need to be reviewed and analyzed to highlight the difference and innovation of this work, such as Nano Energy, 118, 2023, 108997.

4.      How is the pressure sensitivity evaluation of sensors carried out? Why is there a lack of corresponding dynamic response and recovery curves under different standard pressures

5.      As a research paper, the piezoelectric mechanism needs to be provided.

6.      It is recommended to first evaluate the pressure-sensitive characteristics of the sensor before discussing the application of respiratory detection (Figure 5).

7.      Table 2: The table format should be a three-line table.

8.      Most of the references are out of date. It is suggested that references should be concentrated in the last three years.

9.      English writing of the manuscript needs further polishing.

Comments on the Quality of English Language

 Minor editing of English language required.

Author Response

We would like to thank the reviewer for his precious suggestions necessary for improving the quality of our work. All the modifications inserted in the paper file have been green-evidenced to make more visible the reworked parts.

In this paper (sensors-2915828), the authors presented a piezoelectric pressure sensor based on aluminum nitride (AlN) thin film. The results are acceptable and the topic can attract a wide range of readerships. But there are many problems in the introduction, presentation, and discussion of the results. As such, a Major Revision is needed before possible publication. My specific comments are as follows:

  1. Title: “biocompatible” is not necessary to emphasize, and “AlN” needs to be spelled in full in the title.

As rightly observed by the referee, the biocompatibility of AlN has been emphasized in the title without providing a discussion about the importance of this film property for the aim of the work. As a matter of fact, biocompatibility is a mandatory property for wearable technology, and the authors have considered interesting to report that the proposed material meets this requirement. However, in the discussed application it is not used in contact with skin; therefore, we changed the paper title by eliminating the word “biocompatible”, following the reviewer’s suggestion. In addition, AlN has been spelled in full in the title.

  1. Author affiliation: The same affiliation does not need to be listed repeatedly.

The affiliation has been considered just once for all authors, as rightly suggested by the referee.

  1. Introduction: (1) The abbreviations that first appear need to be spelled out in full, such as AIN “…fabrication, AlN is…”. (2) The background logic is a bit confusing. The story should be enhanced. The research on respiratory detection sensors needs to be summarized, such as pressure, strain, and humidity sensors. The more recent research progresses of advanced flexible pressure and respiration sensors need to be reviewed and analyzed to highlight the difference and innovation of this work, such as Nano Energy, 118, 2023, 108997.

The abbreviations were written when they were first mentioned in the introductory paragraph. We have tried to improve the organization of the introduction. Regarding the research on respiratory sensing sensors, as appropriately pointed out by the referee, it was the authors' choice to limit the discussion to only flexible piezoelectric pressure sensors used for monitoring respiration. For this reason, Table 2 provides a comparison of the performance between the piezoelectric pressure sensor proposed in our paper and works from the more recent literature, referring exclusively to flexible piezoelectric pressure sensors used for respiration monitoring, where the deformation of the active area of the sensor is due to respiratory flow. Although it would be interesting and more complete to discuss pressure/deformation/humidity sensors, the authors limited the discussion in this way because the field of flexible pressure sensors and their applications is so wide that the introductory framework would be more desultory.

  1. How is the pressure sensitivity evaluation of sensors carried out? Why is there a lack of corresponding dynamic response and recovery curves under different standard pressures

As described in Figure 5 (of the revised manuscript) the sensitivity of the sensor (calibration curve) is evaluated by recording the response of the sensor at different pressures reported in x-axis of the above-mentioned figure. The output electrical signal generated by the fabricated piezoelectric sensor is acquired by a voltage amplifier (DLPVA-100-F Series - FEMTO), stored via a digital oscilloscope (GDS-2204A), and then processed in the programming language and numeric computing environment software Matlab. The sensitivity of the sensor is given by the linear regression of the output voltage vs applied pressure plot. As rightly noted by the referee, the dynamic response of the sensor does not correspond to recovery curve. The authors attempted to give an explanation in the paper, reporting that the highlighted behaviour could be attributed to the not fully optimized experimental conditions, where the emptying of the controlled atmosphere cell during the simulated breathing act is not so fast, causing the membrane deformation persistence.

  1. As a research paper, the piezoelectric mechanism needs to be provided.

       As suggested by the referee, a brief description was added in the introductory paragraph regarding the piezoelectric mechanism.

  1. It is recommended to first evaluate the pressure-sensitive characteristics of the sensor before discussing the application of respiratory detection (Figure 5).

      Many thanks to the referee for this suggestion which improves the setting of the paper. as correctly suggested, it is better to first evaluate the sensitivity of the sensor as a function of the pressure, and then discuss its use for breath monitoring. For this reason, the Figure 5 was replaced by Figure 7 where the sensitivity of the sensor is evaluated in the respiration pressure range.

  1. Table 2: The table format should be a three-line table.

As recommended by the referee, Table 2 format was modified.

  1. Most of the references are out of date. It is suggested that references should be concentrated in the last three years.

Some references out of date have been replaced in the revised manuscript. Others have been kept because they represent milestones for the discussed topic.

  1. English writing of the manuscript needs further polishing.

English was improved.

Round 2

Reviewer 3 Report

Comments and Suggestions for Authors

There are still some unresolved issues:

1.         Title: The “Wearable piezoelectric” seems too wide, and the authors only verified respiratory rate detection.

2.         Piezoresistive sensors: “slow response to dynamic pressure”, Why?

3.         “temperature dependence offset”. Are other pressure sensors not affected by temperature? Compared to temperature, the pressure-sensitive response is much greater than the influence of temperature. The temperature effect can be ignored.

4.         “low pressures detection and require high voltage operation.” Why? Please follow the example with wide detection range, J. Mater. Chem. C, 2021, 9, 13659–13667.

5.         Even without considering other types of respiratory rate detection sensors, limited to the field of pressure sensors, what is the progress of research on respiratory rate detection? What is the innovation and significance of this work? Attempting to solve what problem in this field.

6.         Figure 4, 5 and 9: To provide persuasive results, can the authors provide dynamic response recovery curves (similar to respiratory rate curves) corresponding to these point data?

7.         The table needs to use a three-line table.

8.         Check the format of the references. Some of the literature information is incomplete, and the journal name needs to be abbreviated.

Comments on the Quality of English Language

Minor editing of English language required.
